# Long-Term Weight Regain Following Bariatric Surgery: The Role of Diet and Eating Behaviors in Saudi Arabia

**DOI:** 10.3390/nu17193080

**Published:** 2025-09-27

**Authors:** Nora A. Althumiri, Nasser F. Bindhim, Abrar Turki, Saja A. Al-Rayes, Arwa Alumran

**Affiliations:** 1Health Information Management and Technology Department, College of Public Health, Imam Abdulrahman bin Faisal University, Dammam 31451, Eastern Province, Saudi Arabia; 2Informed Decision Making (IDM), Riyadh 13303, Saudi Arabia; 3Sharik Association for Research and Studies, Riyadh 13302, Saudi Arabia; 4Clinical Nutrition Department, College of Applied Medical Sciences, University of Hafr Al Batin, Hafr Al Batin 39953, Eastern Province, Saudi Arabia

**Keywords:** weight regain, bariatric surgery, dietary behaviors, weight regain, Saudi Arabia

## Abstract

Background: While many factors contribute to weight regain after bariatric surgery, the role of dietary intake and behaviors remains underexplored in specific populations. This study aims to explore the long-term outcomes of bariatric surgery in Saudi Arabia, focusing on the impact of dietary intake and eating behaviors on weight regain. Methods: A cross-sectional study was conducted using phone interviews to assess the dietary patterns and eating behaviors of 335 participants who underwent bariatric surgery at least three years prior. Data were collected through a structured validated questionnaire covering food consumption and eating habits. Results: Of the 335 participants who completed the survey, 81.8% experienced weight regain, defined as a ≥20% increase from their nadir weight, after bariatric surgery. Several dietary behaviors were associated with weight regain. Higher consumption of pastries (OR = 0.382, *p* = 0.014), sweets (OR = 0.498, *p* = 0.036), and moderate bread intake (OR = 0.287, *p* = 0.038) were associated with lower odds of obesity. Additionally, frequent consumption of traditional dishes such as kabsah (OR = 0.411, *p* = 0.008) was associated with lower odds of obesity. Conclusions: This study highlights the complexity of dietary patterns in post-bariatric surgery weight management. Although certain foods, such as pastries, sweets, and kabsah, were associated with lower odds of obesity in this sample, the cross-sectional design prevents causal interpretation. These findings should therefore be viewed as exploratory signals, emphasizing the need for cautious interpretation and reinforcing the call for longitudinal, representative studies to clarify the long-term determinants of weight regain and to guide clinical follow-up strategies.

## 1. Introduction

Weight regain after bariatric surgery is a well-recognized phenomenon. According to a systematic review, more than 60% of participants experience weight regain within 3 to 10 years following surgery. This weight regain is also associated with the recurrence of comorbidities such as type 2 diabetes, hypertension, and obesity [1]. However, there is no unified definition of weight regain. The most common definition is an increase of 10 kg from the nadir weight [2,3]. Alternative definitions, such as those based on body mass index (BMI), percent excess weight loss (EWL), or total weight change, may be more clinically meaningful and useful for guiding the choice of revisional procedures [1].

It is important to differentiate between insufficient weight loss and weight regain. A clear distinction must be made between two types of weight loss failure: insufficient weight loss, which occurs 18 months after surgery, and progressive weight regain, which happens after initially achieving successful weight loss [1]. Another systematic review revealed that at least 1 in 6 patients experience a  ≥10% weight regain after bariatric surgery [4]. The high prevalence of weight regain has led to a significant increase in revisional bariatric surgeries, which, unfortunately, comes with increased surgical risks and adverse outcomes for the patient [5].

There are many factors that contribute to weight regain after bariatric surgery. Some are modifiable, such as diet nonadherence, psychological, and behavioral factors, while others are non-modifiable, including genetic, hormonal, surgery-related, and environmental factors [6]. Risk factors for weight regain can be classified into five categories: anatomical, genetic, dietary, psychiatric, and temporal. Specifically, factors such as gastrojejunal stoma diameter, gastric volume after sleeve gastrectomy, anxiety, time since surgery, sweet consumption, emotional eating, portion size, food urges, binge eating, and loss of control/disinhibition during eating have been positively associated with weight regain [5,6,7]. On the other hand, factors such as postprandial GLP-1 levels, eagerness to adopt physical activity, self-esteem, social support, fruit and zinc consumption, high-density lipoprotein cholesterol (HDL) levels, and quality of life have been negatively associated with weight regain [4].

One of the key factors contributing to weight regain after bariatric surgery is related to eating behaviors and food consumption. These behaviors include grazing, loss of control while eating, night eating, eating while watching TV, emotional eating, consumption of high-calorie, processed foods, and inadequate protein intake [6,8,9,10]. A systematic review revealed that most studies, predominantly observational, indicated a reduction in energy intake following surgery, as well as inadequate intakes of micronutrients and protein, coupled with excessive fat consumption [11]. The review also highlighted evidence of nutrient imbalances, poor adherence to multivitamin and mineral supplementation, and limited long-term follow-up of patients [11]. Another study found a significant association between grazing behavior and weight regain after bariatric surgery, regardless of the type of surgery or the specific context of grazing [12].

To the best of our knowledge, this is the most recent study conducted in Saudi Arabia that investigates the dietary factors contributing to weight regain after bariatric surgery. In our previous work, we have studied the association of behavioral lifestyle and psychological factors and their influence on weight regain and recurrence of obesity after the bariatric surgery. However, in this study, we aim to explore the long-term outcomes of bariatric surgery among participants in Saudi Arabia, with a specific focus on weight regain, dietary intake, and eating behaviors as they relate to the recurrence of obesity.

## 2. Method

### 2.1. Study Design

This study employed a cross-sectional design using phone interviews conducted between July and August 2024.

### 2.2. Sampling and Participant Recruitment

This study used a convenience sampling technique using Sharik participant database. The Sharik Association for Research and Studies maintains a comprehensive database of individuals who have expressed willingness and provided informed consent to participate in future research studies [13]. This database includes over 250,000 adult participants (aged 18 years and older) from both genders and across all 13 administrative regions of Saudi Arabia. Individuals who had previously reported undergoing bariatric surgery were identified, resulting in a subsample of 1022 eligible participants for this study [13]. These individuals were subsequently contacted to confirm their interest in participating, with each being called up to three times [13].

### 2.3. The Eligibility Criteria

The eligibility criteria for this study required participants to be Saudi residents, aged ≥ 18 years and Arabic speakers. To ensure that weight trajectories were assessed after the initial post-operative period, only participants who had undergone bariatric surgery at least three years prior to data collection were eligible for inclusion in this study. Those who met the criteria were contacted by phone to obtain verbal consent for participation. If no response was received after three contact attempts, the participant was excluded from the target sample.

### 2.4. Questionnaire Development

The questionnaire used in this research was previously developed and validated in a previous paper. It underwent a rigorous validation process to ensure its content validity, clarity, and relevance to the target population. The questionnaire underwent pilot testing with 44 participants from the Sharik database to assess clarity, comprehension, and internal consistency. Expert review by public health specialists and methodologists ensured content validity. Although test–retest reliability could not be conducted due to the cross-sectional design, the pilot phase confirmed the instrument’s usability and stability. The same tool has been applied in multiple national health studies, supporting its robustness in capturing health-related behaviors, including bariatric surgery outcomes.

Dietary Consumption: Participants were asked to provide detailed information about their dietary habits, particularly their intake of raw and cooked foods, including fruits, vegetables, and proteins such as red meat, chicken, egg, fish, and seafood. Other food groups, such as rice and dairy products, were also assessed. Additionally, the questionnaire explored participants’ consumption of fast food and prepared meals, including commonly consumed items like burgers, pizza, pasta, as well as traditional dishes such as kabsah (rice with meat or chicken) and shawarma. Beverage consumption was another area of focus, covering natural and canned juices, soft drinks, energy drinks, and various types of coffee (Saudi coffee and black coffee), along with tea. The frequency of daily main meals and snacks was also recorded to provide a comprehensive view of participants’ dietary patterns.Eating Behaviors: Eating behaviors were assessed through 22 statements designed to capture a wide range of eating habits. These included behaviors such as night eating, passive eating, and emotional eating, along with habits like eating while watching TV, eating alone, or in social settings. The questionnaire also inquired about snacking between meals, overconsumption of sweets or salty foods, controlled eating practices, calorie monitoring, and any specific dietary restrictions. These questions aimed to provide a thorough understanding of participants’ eating behaviors and their potential impact on dietary choices.

### 2.5. Data Collection

Data collection and storage were managed through the ZDataCloud system, which is specifically designed to ensure data quality and integrity through automated processes (e.g., preventing missing fields, filtering invalid entries, and monitoring data flow) while minimizing the need for human intervention. This system reduces data entry errors, though it does not eliminate sampling bias [14]. The system continuously monitors the quality of each data entry, preventing random or invalid submissions not associated with the participants in the study [14]. To ensure data integrity, all questions in the survey were required to be answered, resulting in a complete dataset with no missing data [14]. Each interview lasted between 15 and 20 min. Participants who did not complete the survey were excluded from the final dataset.

During telephone interviews, participants were asked to report their pre-surgery weight, lowest post-surgery (nadir) weight, and current weight at the time of the interview, along with their height. Height was reported once and assumed constant. BMI values were then calculated using these reported measures.

### 2.6. Data Analysis

In this study, weight regain was defined as a ≥20% increase in body weight from the nadir (lowest post-surgery) weight. This threshold was selected to capture clinically meaningful regain, consistent with prior literature and to avoid misclassification due to minor weight fluctuations [15,16].

Frequencies and percentages were used to describe the variables in this study. Logistic regression analysis was employed to examine the associations between dietary intake, eating behaviors, and the likelihood of being classified as having obesity. Both crude and adjusted models (adjusted for age and gender) were used, and results were reported as odds ratios (OR) with 95% confidence intervals (CIs). Statistical significance was set at a *p*-value of <0.05. Data management and analysis were conducted using the Statistical Package for Social Sciences (SPSS, Armonk, NY, USA) [17].

### 2.7. Ethical Considerations

This study was approved by the ethics committee of Imam Abdulrahman bin Faisal University (Approval no. IRB-OGS-2024-03-352), in compliance with national research ethics regulations (registration number: HAP-05-D-003). Verbal informed consent was obtained from participants during the phone interviews and systematically documented in the data collection system. This method ensured that participants were fully informed, and their consent was appropriately captured, aligning with the study’s protocol. Furthermore, this research adheres to the ethical guidelines set forth in the Declaration of Helsinki [18].

## 3. Results

### 3.1. Participant Characteristics

From the Sharik database, 1022 individuals who had previously self-reported undergoing bariatric surgery were initially identified as potential participants. Following the application of the study’s inclusion criteria—which required participants to be Saudi residents aged 18 years or older and to have undergone bariatric surgery at least three years prior—520 individuals were deemed eligible. Of these, a total of 335 participants completed the survey, yielding a response rate of 64.4% among eligible respondents. The final sample comprised 180 females and 155 males, with a mean age of 38.54 years (SD = 12.43), ranging from 18 to 80 years.

Of the participants, 69.3% reported undergoing gastric sleeve surgery, 6.9% reported Roux-en-Y gastric bypass, while 23.9% selected ‘I don’t know’ regarding their procedure type. These responses were retained to maintain transparency and avoid introducing misclassification bias. Notably, after calculating post-surgery, nadir, and current BMI from self-reported height and weight, 81.8% of participants met the study definition of weight regain, i.e., an increase of ≥20% from nadir weight. Additionally, 12.2% underwent revisional bariatric surgery, and 31.0% were using weight loss injections during the study. Table 1 provides further details on participant characteristics.

### 3.2. Eating Intake Patterns

Most participants reported low fruit and vegetable intake, with fewer than 1.8% meeting acceptable levels. Protein intake was also limited, with only about one-third reporting adequate consumption of chicken or red meat and less than one-quarter for fish. By contrast, the majority (83%) consumed eight or more protein-based meals per week when all sources were combined. Carbohydrate intake was high, with 80% consuming eight or more carb-based meals weekly, and rice being a frequent staple. Fast food consumption was also common, with half of the participants reporting eight or more fast food meals per week. Among traditional and Western dishes, kabsah, shawarma, burgers, pizza, and pasta were the most frequently ordered foods. Table 2 showed participants dietary intake patterns. 

### 3.3. Beverage Consumption Patterns

Milk and milk products were consumed regularly, with most participants drinking them at least once a week. Natural and canned juices were moderately consumed, while carbonated and sweetened drinks were common, with nearly half of participants reporting frequent intake. Caffeinated beverages such as black coffee, Saudi coffee, and tea were highly prevalent, with more than 40% consuming them four or more times weekly. Table 3 shows participants’ beverage consumption patterns.

### 3.4. Eating Behavior Patterns

Breakfast was eaten regularly by less than half of participants, and most reported having two to three meals per day. Daily snacking was widespread, with two-thirds reporting at least one snack daily. Unhealthy behaviors were also common, including eating while watching TV (42%) and night eating (35%). Only a small minority (9%) reported monitoring their calorie intake. Table 4 shows participants’ eating behavior patterns.

### 3.5. Factors Influencing Dietary Consumption

In this logistic regression model, several variables showed significant associations with obesity. Gender was significantly associated with lower odds of obesity for females compared to males (OR = 0.475, *p* = 0.007), indicating that females had 52.5% lower odds of being obese. High consumption of pastries (OR = 0.382, *p* = 0.014) and sweets (OR = 0.498, *p* = 0.036) was also associated with reduced odds of obesity, suggesting that individuals who consumed more pastries and sweets had lower odds of being obese. Bread consumption was a significant factor, with those consuming it at moderate levels having 71.3% lower odds of obesity (OR = 0.287, *p* = 0.038) compared to non-consumers. Additionally, frequent consumption of Kabsah (OR = 0.411, *p* = 0.008) was associated with lower odds of obesity, with those consuming it regularly having 58.9% lower odds of being obese. Other variables, including the consumption of shawarma, pizza, or pasta, carbohydrate consumption per week, fast food consumption, sweetened drinks, caffeine intake, or protein consumption as well as various beverages like coffee, tea, juice, and soda were not statistically significant predictors of obesity in this model. However, some trends were observed, such as carbohydrate consumption showing a non-significant protective trend (OR = 0.182, *p* = 0.072).

## 4. Discussion

The aim of this study was to examine the long-term outcomes of bariatric surgery among participants from Saudi Arabia, with a particular focus on weight regain and the influence of dietary intake and behaviors on the recurrence of obesity. The findings revealed significant associations between certain dietary behaviors and obesity in individuals who had undergone bariatric surgery. Females had lower odds of obesity compared to males, while higher consumption of pastries, sweets, and moderate bread intake was linked to a reduced risk of obesity. Additionally, frequent consumption of kabsah was associated with lower odds of obesity. However, other factors such as fast food, carbohydrate intake, and various beverages did not demonstrate statistically significant associations with obesity, indicating that specific dietary patterns may be associated with post-surgery weight management outcomes.

Although our models suggested that higher consumption of pastries, sweets, and kabsah was associated with lower odds of current obesity, these results should be interpreted with great caution. These counterintuitive findings are most likely explained by residual confounding, reporting bias, and the absence of portion size or caloric intake data rather than true protective effects. Given these limitations, we consider these associations to be exploratory signals only and not evidence of a causal or clinically protective relationship. Future studies with detailed dietary assessment, portion size measurement, and longitudinal designs are needed to validate or refute these patterns.

The metabolic changes induced by bariatric surgery, such as improved insulin sensitivity and altered hormone levels, may also affect how the body processes these foods, which could make their impact on weight outcomes appear less significant in observational analysis [19,20,21]. Furthermore, the surgery may lead to reduced nutrient absorption, especially in procedures like gastric bypass, meaning fewer calories from these foods are absorbed [22,23,24]. Moreover, the relationship between increased sweet consumption after bariatric surgery may be influenced by changes in taste sensitivity following the procedure [25,26,27]. While the exact effects of bariatric surgery on taste remain uncertain, some studies suggest that patients, particularly those who undergo Roux-en-Y gastric bypass (RYGB) or vertical sleeve gastrectomy (VSG), may experience heightened sensitivity to sweet flavors [25,28]. This increased sensitivity could result in a stronger preference for sweet foods. Consequently, while patients may consume sweets, their overall caloric intake may remain controlled due to smaller portion sizes and other metabolic changes, which may help explain the observed associations with lower odds of obesity in this sample. However, it is important to note that, due to the study design, we measured the frequency of consumption rather than portion size, which may have influenced the interpretation of the results.

The lack of a significant association between dietary intake or dietary behaviors and obesity, despite the fact that over 80% of individuals who undergo bariatric surgery experience some degree of weight regain and over 30% are reclassified as obese, may be explained by several interrelated factors. First, bariatric surgery induces profound metabolic and physiological changes that affect weight regulation beyond simple dietary intake or behaviors. Hormonal shifts, including changes in ghrelin, leptin, and GLP-1 levels, play a role in influencing hunger, satiety, and energy expenditure [21,29]. These changes may lead to weight regain regardless of an individual’s dietary habits. As a result, weight regain may occur independently of specific dietary patterns, as the altered metabolic environment post-surgery could influence outcomes beyond dietary behaviors alone.

Additionally, weight regain after bariatric surgery is a multifactorial process involving more than just dietary factors [5]. Psychological, behavioral, genetic, and environmental factors all contribute to long-term weight management. For example, stress, emotional eating, or reduced physical activity levels can promote weight regain even when individuals maintain healthy eating habits [30,31,32]. In some cases, weight regain may be driven by psychological or metabolic issues that are not directly tied to dietary intake. Furthermore, many patients undergo behavioral adaptations after surgery that may not be accurately reflected in traditional dietary assessments. For example, patients may engage in grazing, or frequent snacking on small amounts of high-calorie foods, which may be associated with weight regain but are not always captured in standard dietary behavior questionnaires. Bariatric patients also often experience changes in food preferences and portion control, which may obscure the relationship between dietary behavior and weight outcomes in statistical analysis [7,33]. Given the cross-sectional design, our results reflect associations at a single point in time and should not be interpreted as evidence of causal relationships between dietary behaviors and weight regain.

The limitations of this study should be acknowledged to provide appropriate context for the findings. First, all data on dietary intake and eating behaviors were self-reported, which introduces the risk of recall bias and social desirability bias. Participants may have underreported unhealthy food consumption or overreported healthier behaviors, leading to potential misclassification. Similarly, BMI values were derived from self-reported weight and height, which may also be affected by recall bias, although this approach is commonly used in public health research.

Second, while our eligibility criteria required participants to have undergone bariatric surgery at least three years before the survey, precise dates of surgery were not consistently available, preventing the inclusion of “time since surgery” as a continuous covariate in regression models. In addition, a notable proportion of participants (23.9%) were unable to specify their exact surgery type and selected “I don’t know,” which reduces the precision of procedure-specific analyses.

Third, although the questionnaire was validated through expert review and pilot testing and has been successfully used in prior national studies, measurement error cannot be entirely ruled out. Psychometric testing such as test–retest reliability was not feasible within the cross-sectional design. Fourth, our operational definition of weight regain (≥20% increase from nadir weight) was more conservative than the ≥10% threshold commonly used in other studies. This approach ensured that our prevalence estimates reflected clinically meaningful regain but may limit comparability with some published literature.

Finally, the dietary data were based on frequency of food consumption rather than portion sizes or caloric intake, which constrains our ability to estimate true energy intake or the quantitative impact of diet on weight outcomes. Combined with the cross-sectional design, which captures data at a single timepoint, this limits our ability to draw causal inferences or assess changes in behaviors and weight trajectories over time.

Weight regain after bariatric surgery is a dynamic process that evolves over time, and longitudinal studies are needed to better assess the long-term associations between dietary behaviors and weight outcomes.

## 5. Conclusions

In conclusion, this study explored dietary intake and eating behaviors among Saudi patients at least three years after bariatric surgery. Although unexpected associations were observed between certain foods and obesity, these findings should be regarded as exploratory signals only, as they are likely influenced by residual confounding, reporting bias, and the absence of portion size data. The results do not provide evidence for causal mechanisms and should not be interpreted as protective effects.

Importantly, the study design—cross-sectional, self-reported, and based on a volunteer registry—does not allow for causal inference or population-level generalization. Nevertheless, the high prevalence of weight regain observed highlights the need for continued clinical monitoring and underscores the importance of conducting longitudinal studies with more comprehensive clinical and behavioral data to clarify the long-term drivers of post-surgery outcomes.

These findings represent exploratory associations and cannot establish causality; longer follow-up periods and comprehensive longitudinal studies are needed to identify the true drivers of long-term weight regain following bariatric surgery.

## Figures and Tables

**Table 1 nutrients-17-03080-t001:** Overview of Participant Demographics and Health Metrics.

Participant Characteristics	*n* (%)
Surgery Type	
Gastric Sleeve Surgery	232 (69.3)
I don’t know	80 (23.9)
Roux-en-Y gastric bypass	23 (6.9)
Prevalence of Weight Regain. Change in nadir weight to current weight	274 (81.8)
Revisional Bariatric Surgery (Yes)	41 (12.2)
Currently Using Weight Loss Injection (Yes)	104 (31.0)
Health Self-Rate	
Great	85 (25.4)
Very Good	112 (33.4)
Good	79 (23.6)
Fair	33 (9.9)
Poor	26 (7.8)

**Table 2 nutrients-17-03080-t002:** Dietary Intake Patterns of Participants.

Dietary Consumption	*n* (%)
Raw and Cooked Food	
Vegetables	
Acceptable Vegetable level	34 (10.1)
Not Acceptable Vegetable level	301 (89.9)
Fruits	
Acceptable Fruit level	23 (6.9)
Not Acceptable Fruit level	312 (93.1)
Vegetables and Fruits	
Acceptable Vegetable and Fruit level	6 (1.8)
Not Acceptable Vegetable and Fruit level	329 (98.2)
Fish	
Acceptable Fish level	80 (23.9)
Not Acceptable Fish level	255 (76.1)
Red Meat	
Acceptable Red Meat level	122 (36.4)
Not Acceptable Red Meat level	213 (63.6)
Chicken	
Acceptable Chicken level	120 (35.8)
Not Acceptable Chicken level	215 (64.2)
Egg	
No	68 (20.3)
1–3/weeks	139 (41.5)
4 or more	128 (38.2)
Consumption of Protein Per Week	
Never	17 (5.1)
1–3 protein meals/week	7 (2.1)
4–7 protein meals/week	33 (9.9)
8 or more protein meals/week	268 (83.0)
Bread	
No	40 (11.9)
1–3/weeks	118 (35.2)
4 or more	177 (52.8)
Rice	
Never	18 (5.4)
1–7 carb meals/week	167 (49.9)
8 or more carb meals/week	150 (44.8)
Consumption of Carbs Per Week	
Never	7 (2.1)
1–3 carb meals/week	14 (4.2)
4–7 carb meals/week	45 (13.4)
8 or more carb meals/week	269 (80.3)
Most Ordered Food	
Shawarma	
No	137 (40.9)
1–3/weeks	138 (41.2)
4 or more	60 (17.9)
Kabsah (Rice with meat or chicken)	
No	56 (16.7)
1–3/weeks	169 (50.4)
4 or more	110 (32.8)
Burger	
No	115 (34.3)
1–3/weeks	158 (47.2)
4 or more	62 (18.5)
Pizza	
No	138 (41.2)
1–3/weeks	147 (43.9)
4 or more	50 (14.9)
Pasta	
No	130 (38.8)
1–3/weeks	142 (42.4)
4 or more	63 (18.8)
Consumption of Order Fast Food Per Week	
Never	12 (3.6)
1–3 fast food meals/week	50 (14.9)
4–7 fast food meals/week	103 (30.7)
8 or more fast food meals/week	170 (50.7)

1. All variables were based on self-reported consumption frequencies obtained during structured telephone interviews. 2. Food and beverage intake was reported as the number of times per week each item or group was consumed. Categories were collapsed into uniform ranges (e.g., none, 1–3 times, 4–7 times, ≥8 times per week) for clarity. 3. “Acceptable” intake levels for fruits, vegetables, and proteins were defined according to national and international dietary guidelines, while “not acceptable” reflects intake below those thresholds. 4. “Protein meals per week” was calculated by summing self-reported weekly consumption of chicken, red meat, fish, eggs, and other protein-rich foods. 5. “Carbohydrate meals per week” included rice, bread, pasta, and other major carbohydrate sources. 6. Fast food items included burgers, pizza, pasta, shawarma, and traditional rice-based dishes such as kabsah. 7. All percentages were calculated based on the total sample (n = 335). Due to rounding, percentages may not sum to exactly 100%.

**Table 3 nutrients-17-03080-t003:** Beverage Consumption Patterns.

Beverages	*n* (%)
Milk and milk products	
No	50 (14.9)
1–3/weeks	134 (40.0)
4 or more	151 (45.1)
Natural Juice	
Never	120 (35.8)
1–3 times/week	149 (44.5)
4–7 times/week	66 (19.7)
Canned Juice	
Never	130 (38.1)
1–3 times/week	126 (37.6)
4–7 times/week	79 (23.6)
Carbonated Drinks	
Never	109 (32.5)
1–3 times/week	130 (38.8)
4–7 times/week	96 (28.7)
Power Drinks	
Never	204 (60.9)
1–3 times/week	80 (23.9)
4–7 times/week	51 (15.2)
Consumption of Sweated Drinks Per Week	
Never	32 (9.6)
1–3 drinks/week	67 (20.0)
4–7 drinks/week	86 (25.7)
8 drinks/week	150 (44.8)
Black Coffee	
No	91 (27.2)
1–3/week	123 (36.7)
4 or more	121 (36.1)
Saudi Coffee	
No	69 (20.6)
1–3/week	124 (37.0)
4 or more	142 (42.4)
Caffeine Drinks Per Week	
Never	32 (9.6)
1–3 drinks/week	67 (20.0)
4–7 drinks/week	86 (25.6)
8 drinks/week	150 (44.8)
Tea	
No	66 (19.7)
1–3/week	123 (36.7)
4 or more	146 (43.6)

1. All variables were based on self-reported consumption frequencies obtained during structured telephone interviews. 2. Food and beverage intake was reported as the number of times per week each item or group was consumed. Categories were collapsed into uniform ranges (e.g., none, 1–3 times, 4–7 times, ≥8 times per week) for clarity. 3. Caffeinated drinks included black coffee, Saudi coffee, and tea. Sweetened beverages included sodas, canned juices, and energy drinks. 4. All percentages were calculated based on the total sample (n = 335). Due to rounding, percentages may not sum to exactly 100%.

**Table 4 nutrients-17-03080-t004:** Eating Behavior Patterns.

Eating Behaviors		*n* (%)
Breakfast		
Never	63 (18.8)	
1–3 days per week	126 (37.6)	
4–7 days per week	146 (43.6)	
Number of meals per day		
1 meal	75 (22.4)	
2–3 meals	192 (57.3)	
more than 3	68 (20.3)	
Number of Snacks		
Nothing	73 (21.8)	
1 snack/day	225 (67.2)	
2 or more/day	37 (11.0)	
High Consumption of pastry—Yes	50 (14.9)	
High Consumption of Sweets—Yes	90 (26.9)	
Eating While Watching TV—Yes	141 (42.1)	
Night Eating—Yes	118 (35.2)	
Calorie Monitoring Behavior—Yes	29 (8.7)	
Social Dining Frequency—Yes	71 (21.2)	

1. All variables were based on self-reported consumption frequencies obtained during structured telephone interviews. 2. Food and beverage intake was reported as the number of times per week each item or group was consumed. Categories were collapsed into uniform ranges (e.g., none, 1–3 times, 4–7 times, ≥8 times per week) for clarity. 3. All percentages were calculated based on the total sample (n = 335). Due to rounding, percentages may not sum to exactly 100%.

## Data Availability

The data are available and can be requested through email: na@idm.sa.

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
