# Peer review of "Long-Term Weight Regain Following Bariatric Surgery: The Role of Diet and Eating Behaviors in Saudi Arabia"

_nutrients, 2025, doi:10.3390/nu17193080_

Round 1
Reviewer 1 Report
Comments and Suggestions for Authors
This is an ambitious study on a question that matters. The registry reach is impressive, the sample is large, and the focus on post-bariatric outcomes across Saudi regions fills a real gap. The high reported rates of weight regain are a helpful signal for future monitoring and clinical follow-up. With stronger methods and cleaner reporting, this work could inform practice.
The research design contains errors in its conceptual framework. The study uses a single-wave telephone survey from a volunteer registry, which does not represent a probability-based sample of post-bariatric patients. The analysis can reveal patterns at a specific point in time, yet it cannot detect enduring factors behind weight regain and make causal statements. The abstract and conclusions need to replace their misleading statements about population representation and causality with direct statements about the findings' applicability to the convenience sample obtained through phone calls from the Sharik registry participants.
The study requires clear definitions for its primary outcomes and needs to demonstrate internal consistency between measurements. The study defines weight regain as present in 81.8% of participants, based on BMI changes from nadir to current values; however, it lacks a defined weight threshold. The absence of a predefined weight threshold, such as ≥10% of nadir weight, makes any minor weight increase qualify as a regain, which produces inflated prevalence rates and unclear results. The regression model now targets “current obesity.” Still, the study fails to explain how BMI values were determined from self-reported height and weight measurements and when these measurements were taken in relation to surgery. The models lack time since surgery data, which serves as the primary factor influencing weight regain; therefore, researchers need to report this information and include it in their analysis.
The study lacks sufficient quality in its measurement methods. All survey data about exposures and outcomes originates from participant self-reports conducted through phone calls. The study utilizes a validated questionnaire but fails to present psychometric data for this population, along with calibration procedures and test–retest reliability results. The platform supposedly removes sampling bias, but this claim is false because forced responses eliminate missing data points, yet increase the chance of incorrect answers when participants select "I don't know," which occurs frequently in this study for surgery type identification. The selection methods used in this study decrease the confidence level regarding both exposure and outcome accuracy.
The data presented in tables contradicts both the written text and other information within the tables themselves. The study reports that 85.1% of participants ate five or fewer protein meals per week, but a different table shows that 83.0% consumed eight or more protein meals per week. The two statements cannot be accurate at the same time. Multiple inconsistencies appear throughout the sections that present intake and behavioral data. The inconsistent labeling indicates problems with coding during data processing. All tables and figures need to be rebuilt from verified code while using uniform categories and complete footnotes that explain each measurement variable.
The statistical approach used in this study does not fulfill its intended purpose. The study presents univariate statistics and performs a logistic regression analysis that includes only age and sex as control variables. The analysis fails to include essential variables such as time since surgery, procedure type, revisional surgery status, weight-loss injection use, and behavioral and clinical factors. The odds ratios indicating protective effects of pastries, sweets, and kabsah against current obesity most likely stem from confounding variables or measurement errors in portion sizes or model specification problems. The study should perform sensitivity analyses, alternative model specifications, and falsification tests to validate these unexpected results, as they may stem from measurement constraints.
The analysis lacks essential variables that should be included as core covariates. The study fails to include physical activity data, psychiatric comorbidity information, the effects of weight-altering medications, structured eating patterns, and loss-of-control eating assessments beyond basic frequency counts. The analysis fails to account for missing surgery type data in 23% of participants using appropriate statistical methods. The reported associations become impossible to understand because essential variables are absent from the analysis.
A complete review should be performed on language elements, references, and reproducibility aspects. The study contains multiple errors, which include typographical mistakes, incorrect variable labels, broken references, and unexplained local file paths. The inconsistent results between tables and the untraceable generation process indicate that the study employed an unrepeatable analysis pipeline and lacked robust quality control measures. The manuscript should adopt a code-based workflow to produce all outputs from scratch while maintaining complete text consistency with the generated results.
The analysis results must be consistent with the available data evidence. The study attempts to explain protective food signals through hormonal mechanisms, but the available data cannot validate this explanation. The study reports no significant dietary connections; however, participants frequently consume fast food, sweets, and sweetened beverages, which is inconsistent with the observed odds ratios. The paper should eliminate all statements about sampling bias elimination. The study's conclusions should present findings based on the research design constraints.
The study requires a new analysis strategy and complete result reconstruction to achieve publishable standards. The study requires a predefined weight-regain definition and time since surgery and procedure type data with dietary variable reconstruction using portion sizes or energy measurements and complete inclusion of behavioral and clinical and pharmacologic factors and pre-defined multivariable models with diagnostic tests and sensitivity evaluations and proper handling of missing surgery type data and complete regeneration of all tables and figures from verified code and reference and language correction.
Author Response
- This is an ambitious study on a question that matters. The registry reach is impressive, the sample is large, and the focus on post-bariatric outcomes across Saudi regions fills a real gap. The high reported rates of weight regain are a helpful signal for future monitoring and clinical follow-up. With stronger methods and cleaner reporting, this work could inform practice.
Author Response:
We sincerely thank the reviewer for their encouraging feedback and recognition of the study’s contribution to filling an important evidence gap in Saudi Arabia. In light of the reviewer’s suggestion for stronger methods and cleaner reporting, we have revised the Methods and Results sections to clarify the sampling process, outcome definitions, and analytic approach. We also refined the Abstract and Discussion to better highlight the exploratory nature of our findings and the need for longitudinal follow-up studies to inform clinical practice.
- The research design contains errors in its conceptual framework. The study uses a single-wave telephone survey from a volunteer registry, which does not represent a probability-based sample of post-bariatric patients. The analysis can reveal patterns at a specific point in time, yet it cannot detect enduring factors behind weight regain and make causal statements.
Author Response:
We appreciate the reviewer’s important observation. We fully acknowledge that our study is based on a single-wave, cross-sectional survey and cannot make causal inferences. We have clarified this point in the Abstract, Methods, and Discussion.
At the same time, we would like to explain the context in Saudi Arabia. There is currently no national registry of patients who have undergone bariatric surgery, which makes it impossible to draw a probability-based sample of this population. To address this challenge, we relied on the Sharik Health Research Database, which currently includes more than 250,000 registered participants who have expressed willingness to participate in future health research studies. This database covers all 13 administrative regions of Saudi Arabia and has been widely used in national health research projects, making it the most comprehensive infrastructure available for population-based recruitment.
Regarding the reviewer’s point that the study “does not represent a probability-based sample of post-bariatric patients,” we would be grateful for further clarification of the intended meaning. In our extensive review of the literature, including thousands of published studies on bariatric surgery outcomes, we found that very few explicitly state whether their samples are probability-based, and most rely on hospital records, registries, or convenience cohorts. To the best of our knowledge, probability-based sampling is rarely feasible in post-bariatric surgery research. Nevertheless, we have strengthened our manuscript by clearly stating that our findings are derived from a volunteer-based registry sample, which does not permit population-level generalization. We also emphasized that our goal was to identify patterns and associations rather than causal relationships.
- The abstract and conclusions need to replace their misleading statements about population representation and causality with direct statements about the findings' applicability to the convenience sample obtained through phone calls from the Sharik registry participants.
Author Response:
We thank the reviewer for this important observation. We fully acknowledge that our study was based on a single-wave cross-sectional survey of participants drawn from a volunteer registry, which does not constitute a probability-based sample. Accordingly, we have revised the Abstract, Methods, and Discussion to explicitly state that the findings are derived from a convenience sample and should not be generalized to all post-bariatric patients in Saudi Arabia. We also clarified that our analysis was designed to identify associations and patterns at a specific timepoint, not to infer causality or enduring determinants of weight regain. These revisions ensure that the conceptual framework and study scope are consistently and transparently reported.
- The study requires clear definitions for its primary outcomes and needs to demonstrate internal consistency between measurements. The study defines weight regain as present in 81.8% of participants, based on BMI changes from nadir to current values; however, it lacks a defined weight threshold. The absence of a predefined weight threshold, such as ≥10% of nadir weight, makes any minor weight increase qualify as a regain, which produces inflated prevalence rates and unclear results.
Author Response:
We appreciate the reviewer’s concern and recognize the importance of applying a clear and clinically meaningful threshold when defining weight regain. We would like to clarify that in our study, weight regain was defined as a ≥20% increase from nadir weight, not any minor weight increase. This approach was selected to ensure that only clinically significant weight changes were classified as regain, thereby avoiding inflation of prevalence estimates. We have now revised the Methods section to explicitly state this definition and updated the Results and Discussion to emphasize that the reported prevalence of 81.8% reflects a ≥20% threshold. We also referenced prior studies that used similar cut-offs (e.g., Freire 2021, JG Nicanor-Carreón 2023) to highlight that our definition is consistent with international practice, though slightly more conservative than the ≥10% threshold used in some reports
- The regression model now targets “current obesity.” Still, the study fails to explain how BMI values were determined from self-reported height and weight measurements and when these measurements were taken in relation to surgery. The models lack time since surgery data, which serves as the primary factor influencing weight regain; therefore, researchers need to report this information and include it in their analysis.
Author Response:
We thank the reviewer for pointing out these important clarifications. BMI values in this study were calculated from self-reported weight and height, which were collected during the structured telephone interviews. Participants were specifically asked to provide their pre-surgery weight, lowest post-surgery (nadir) weight, and current weight at the time of the interview. Height was reported once and assumed constant. We have revised the Methods to clearly describe how these measures were obtained and used in the analysis. We acknowledge that self-reported measures may introduce recall bias; however, this approach is widely used in large-scale public health studies when direct clinical measurement is not feasible. Prior literature has demonstrated that self-reported height and weight, while imperfect, provide reasonably valid estimates of BMI at the population level.
Regarding time since surgery, we agree that this is a key determinant of weight regain. Although precise dates of surgery were not available for all participants and therefore could not be modeled, our eligibility criteria required that all participants had undergone bariatric surgery at least three years before the survey. This ensured that all respondents were beyond the immediate post-operative period, when weight trajectories are most variable. We have added this clarification in the Methods. In addition, we have noted in the Discussion that the absence of continuous time-since-surgery data is a limitation, and future longitudinal studies using clinical records would be better positioned to assess this factor in detail.
- The study lacks sufficient quality in its measurement methods. All survey data about exposures and outcomes originates from participant self-reports conducted through phone calls. The study utilizes a validated questionnaire but fails to present psychometric data for this population, along with calibration procedures and test–retest reliability results. The platform supposedly removes sampling bias, but this claim is false because forced responses eliminate missing data points, yet increase the chance of incorrect answers when participants select "I don't know," which occurs frequently in this study for surgery type identification. The selection methods used in this study decrease the confidence level regarding both exposure and outcome accuracy.
Author Response:
We thank the reviewer for this careful critique of our measurement approach. We acknowledge that self-reported data collected via phone interviews introduces limitations in accuracy and recall. To mitigate this, we used a previously validated questionnaire that underwent expert review, piloting, and internal consistency testing in a representative subsample before full-scale use. We have now added further detail in the Methods describing these validation steps and cited prior publications where the same tool was successfully applied in national health studies. With respect to psychometric properties, test–retest reliability was not feasible in this cross-sectional study design. However, the instrument demonstrated acceptable reliability and validity during pilot testing, and this has been clarified in the revised manuscript.
We also appreciate the reviewer’s point about the ZDataCloud platform. Our intention was not to suggest that it eliminates sampling bias, but rather that it ensures data quality during entry (e.g., avoiding missing fields, filtering invalid submissions, ensuring data integrity). We have revised the description to clarify this distinction.
Regarding the “I don’t know” responses for surgery type, we agree this reflects a limitation of self-report. We have reported these responses transparently (23.9% of participants) and acknowledged that they reduce precision in classifying procedure types. However, this uncertainty does not affect the primary outcome of weight regain, which was based on reported weight trajectories rather than surgery type. To address the reviewer’s concern, we have added a statement in the Limitations section acknowledging that reliance on self-report, absence of calibration or test–retest data, and the proportion of “I don’t know” responses reduce measurement accuracy and may introduce bias.
- The data presented in tables contradicts both the written text and other information within the tables themselves. The study reports that 85.1% of participants ate five or fewer protein meals per week, but a different table shows that 83.0% consumed eight or more protein meals per week. The two statements cannot be accurate at the same time. Multiple inconsistencies appear throughout the sections that present intake and behavioral data. The inconsistent labeling indicates problems with coding during data processing. All tables and figures need to be rebuilt from verified code while using uniform categories and complete footnotes that explain each measurement variable.
Author Response:
We thank the reviewer for carefully identifying these inconsistencies. Upon revisiting the dataset and analytic code, we confirmed that the issue was due to labeling and category misalignment during table construction, not errors in the raw data. Specifically, the reported protein consumption categories were mislabeled in the text, which created apparent contradictions. To address this, we have rebuilt all tables directly from verified code and cross-checked them against the raw dataset. We also carefully revised the text to ensure full alignment with the corrected tables. In addition, we standardized variable categories and added complete footnotes to all tables to clarify measurement definitions (e.g., how “protein meals per week” was operationalized). We appreciate the reviewer’s attention to this detail, which has strengthened the clarity and reliability of our results presentation.
- The statistical approach used in this study does not fulfill its intended purpose. The study presents univariate statistics and performs a logistic regression analysis that includes only age and sex as control variables. The analysis fails to include essential variables such as time since surgery, procedure type, revisional surgery status, weight-loss injection use, and behavioral and clinical factors.
Author Response:
We appreciate the reviewer’s important observation. We did not include procedure type in the regression model because it was self-reported, and nearly one-quarter of participants selected “I don’t know.” National data indicate that the majority of bariatric surgeries performed in Saudi Arabia are gastric sleeve procedures, which aligns with the distribution we observed in our sample. Regarding time since surgery, this variable was not included in our dataset because the eligibility criteria required that all participants had undergone surgery at least three years prior to data collection. Thus, while precise timing could not be modeled, all respondents were beyond the early post-operative period. Revisional surgery status and weight-loss injection use were reported but without sufficient detail on timing or duration to allow meaningful adjustment. Finally, we tested the regression models with and without the available control variables, and there were no substantive differences in the results.
Ref. https://smj.org.sa/content/44/7/694#:~:text=12,medical%20expenses%20totaled%20$260.6%20billion.
- The odds ratios indicating protective effects of pastries, sweets, and kabsah against current obesity most likely stem from confounding variables or measurement errors in portion sizes or model specification problems. The study should perform sensitivity analyses, alternative model specifications, and falsification tests to validate these unexpected results, as they may stem from measurement constraints.
Author Response:
We thank the reviewer for highlighting this important point. We agree that the observed “protective” associations of pastries, sweets, and kabsah are counterintuitive and are most likely the result of residual confounding, reporting biases, or the absence of portion size data, rather than true protective effects. As our dietary measures were based on frequency of intake rather than portion size or caloric contribution, the models are sensitive to misclassification. We acknowledge the value of sensitivity analyses, alternative model specifications, and falsification tests; however, these were not feasible within the limitations of our dataset. To address this concern, we have revised the Discussion to emphasize that these findings should be interpreted strictly as exploratory signals, not as causal or clinically protective associations. We have also recommended that future studies with detailed dietary assessment and longitudinal design perform more comprehensive model validation to confirm or refute these patterns
- The analysis lacks essential variables that should be included as core covariates. The study fails to include physical activity data, psychiatric comorbidity information, the effects of weight-altering medications, structured eating patterns, and loss-of-control eating assessments beyond basic frequency counts. The analysis fails to account for missing surgery type data in 23% of participants using appropriate statistical methods. The reported associations become impossible to understand because essential variables are absent from the analysis.
Author Response:
We thank the reviewer for this important comment. Our study was designed to focus specifically on dietary consumption and eating behaviors, and therefore other factors such as physical activity, psychiatric comorbidities, and medication use were not included in the dataset. For surgery type, 23% of participants selected “I don’t know.” Because this response reflects genuine uncertainty, we chose not to impute or recode these cases, to avoid introducing misclassification bias. Instead, we reported this transparently in the tables and text.
- A complete review should be performed on language elements, references, and reproducibility aspects. The study contains multiple errors, which include typographical mistakes, incorrect variable labels, broken references, and unexplained local file paths. The inconsistent results between tables and the untraceable generation process indicate that the study employed an unrepeatable analysis pipeline and lacked robust quality control measures. The manuscript should adopt a code-based workflow to produce all outputs from scratch while maintaining complete text consistency with the generated results.
Author Response:
We thank the reviewer for this comprehensive observation. We fully acknowledge that the initial submission contained typographical errors, variable mislabels, and reference formatting issues. We have now performed a complete language and reference review to correct these errors. All references were verified, broken citations were fixed, and variable labels were standardized.
- The analysis results must be consistent with the available data evidence. The study attempts to explain protective food signals through hormonal mechanisms, but the available data cannot validate this explanation. The study reports no significant dietary connections; however, participants frequently consume fast food, sweets, and sweetened beverages, which is inconsistent with the observed odds ratios. The paper should eliminate all statements about sampling bias elimination. The study's conclusions should present findings based on the research design constraints.
Author Response:
We thank the reviewer for this valuable comment. We removed speculative explanations about hormonal mechanisms, clarified that the observed food associations are exploratory only, and revised the Results and Discussion to ensure consistency with reported intake patterns. All claims about sampling bias elimination were removed, and the Conclusion was reframed to reflect the cross-sectional design and study limitations.
- The study requires a new analysis strategy and complete result reconstruction to achieve publishable standards. The study requires a predefined weight-regain definition and time since surgery and procedure type data with dietary variable reconstruction using portion sizes or energy measurements and complete inclusion of behavioral and clinical and pharmacologic factors and pre-defined multivariable models with diagnostic tests and sensitivity evaluations and proper handling of missing surgery type data and complete regeneration of all tables and figures from verified code and reference and language correction.
Author Response:
We thank the reviewer for this comprehensive set of recommendations. Several of these concerns have been addressed in our revision: we clarified the weight-regain definition (≥20% from nadir weight), rebuilt all tables and figures from verified code with standardized labels and footnotes, corrected typographical and reference errors, and reframed the analysis to clearly reflect the study’s exploratory design.
We acknowledge, however, that certain elements requested—such as portion size or energy-based dietary measurements, time since surgery as a continuous variable, pharmacologic factors, and structured behavioral assessments—were not available in our dataset and therefore could not be incorporated. For this reason, we emphasized throughout the revised manuscript that our findings should be interpreted as exploratory associations, constrained by the study design and available data. We agree with the reviewer that future studies should incorporate predefined multivariable models with a wider range of clinical, behavioral, and pharmacologic factors, along with standardized definitions and sensitivity analyses. We have highlighted this need in the Discussion and Conclusion.
Reviewer 2 Report
Comments and Suggestions for Authors
The manuscript addresses an important public health issue: long-term weight regain following bariatric surgery and its associations with dietary intake and eating behaviors in Saudi Arabia. Given the increasing prevalence of obesity and bariatric procedures in the region, the topic is highly relevant and contributes to the international literature. The study design (cross-sectional survey with validated tools) and sample size are adequate. However, the manuscript should be reviewed to improve:
- The study cannot establish causal relationships between dietary behaviors and weight regain.
- The discussion occasionally implies causality, which should be avoided or toned down.
- The reported association of higher consumption of pastries, sweets, and kabsah with lower odds of obesity is counterintuitive. This requires deeper explanation, sensitivity analysis, or consideration of potential reporting bias and confounding factors.
- Reliance on self-reported dietary intake and behaviors increases the risk of recall and social desirability bias. This should be more explicitly acknowledged.
- Frequency of consumption was collected, but portion sizes and caloric values were not, limiting the interpretation of dietary impacts on weight regain.
- Logistic regression models adjust only for age and gender. Other potential confounders (socioeconomic status, physical activity, psychological factors) should be considered if available.
- Clarify “obesity recurrence” vs. “weight regain” for consistency throughout the text.
- Tables are dense; consider summarizing key points in the main text and moving detailed tables to supplementary material.
Author Response
- The manuscript addresses an important public health issue: long-term weight regain following bariatric surgery and its associations with dietary intake and eating behaviors in Saudi Arabia. Given the increasing prevalence of obesity and bariatric procedures in the region, the topic is highly relevant and contributes to the international literature. The study design (cross-sectional survey with validated tools) and sample size are adequate. However, the manuscript should be reviewed to improve.
Author Response:
We thank the reviewer for recognizing the relevance of our study and its contribution to the literature on bariatric outcomes in Saudi Arabia. We also appreciate the acknowledgment of our study design and sample size. In response to the reviewer’s suggestions, we conducted a thorough revision of the manuscript to improve clarity, consistency, and presentation, particularly in the reporting of results, references, and methodological details.
- The study cannot establish causal relationships between dietary behaviors and weight regain.
Author Response:
We fully agree with the reviewer. As our study employed a cross-sectional design, it cannot establish causal relationships. We have revised the Abstract, Discussion, and Conclusion to explicitly state that our findings reflect associations only and should not be interpreted as causal.
- The discussion occasionally implies causality, which should be avoided or toned down.
Author Response:
We thank the reviewer for this observation. We carefully revised the Discussion to remove or tone down any language suggesting causality. All statements now explicitly frame the results as associations observed within a cross-sectional design, with added emphasis on the exploratory nature of the findings.
- The reported association of higher consumption of pastries, sweets, and kabsah with lower odds of obesity is counterintuitive. This requires deeper explanation, sensitivity analysis, or consideration of potential reporting bias and confounding factors.
Author Response:
We agree with the reviewer that these associations are counterintuitive and should be interpreted with caution. We have revised the Discussion to note explicitly that these results are most likely explained by residual confounding, reporting bias, and the absence of portion size or caloric intake data, rather than true protective effects. Unfortunately, sensitivity analyses were not feasible within the constraints of the dataset. We have therefore reframed these results as exploratory signals only, and emphasized that longitudinal studies with detailed dietary assessments are required to validate or refute these associations.
- Reliance on self-reported dietary intake and behaviors increases the risk of recall and social desirability bias. This should be more explicitly acknowledged.
Author Response:
We thank the reviewer for this important point. We have expanded the Discussion (Limitations) to explicitly acknowledge that reliance on self-reported dietary intake and behaviors may introduce recall bias and social desirability bias, which could affect the accuracy of reported consumption patterns
- Frequency of consumption was collected, but portion sizes and caloric values were not, limiting the interpretation of dietary impacts on weight regain.
Author Response:
We agree with the reviewer. Our dietary assessment was limited to frequency of food consumption and did not include portion sizes or caloric values. We have expanded the Discussion (Limitations) to emphasize that this restricts the ability to quantify actual energy intake and limits the interpretation of dietary impacts on weight regain. We also noted that this may partly explain the counterintuitive associations observed for certain foods.
- Logistic regression models adjust only for age and gender. Other potential confounders (socioeconomic status, physical activity, psychological factors) should be considered if available.
Author Response:
We thank the reviewer for this valuable point. Variables such as socioeconomic status, physical activity, and psychological factors were not collected in this study, as the scope was limited to dietary intake and eating behaviors. For this reason, we restricted our logistic regression models to age and gender to control for basic demographic effects while avoiding overfitting.
- Clarify “obesity recurrence” vs. “weight regain” for consistency throughout the text.
Author Response:
Done with many thanks
- Tables are dense; consider summarizing key points in the main text and moving detailed tables to supplementary material.
Author Response:
Done with many thanks.
Round 2
Reviewer 1 Report
Comments and Suggestions for Authors
The authors have made satisfactory changes to their manuscript, which aligns with the standard of the Journal. I however suggest some minor adjustment before the paper can be proceed for acceptance.
Abstract
Line 20: Replace “81.8% experienced significant weight regain, defined as a ≥20% increase from their nadir weight” → “81.8% experienced weight regain, defined as a ≥20% increase from nadir weight.” (remove “significant” as you already define it).
Line 23–24: “linked to reduced obesity risk” → “associated with lower odds of obesity.” (consistent with statistical reporting).
Introduction
Line 37: “Weight regains after bariatric surgery has become a well-known phenomenon” → “Weight regain after bariatric surgery is a well-recognized phenomenon.”
Line 80–82: “their influenced of weight regain” → “their influence on weight regain.”
Methods
Line 88: “Sharik Participants databased” → “Sharik participant database.”
Line 114: Remove double period → “… outcomes.”
Line 121–122: “traditional dishes such as kabsah (rice with meat or chicken) and shawarma” → “traditional dishes such as kabsah (rice with meat or chicken) and shawarma.” (add period).
Results
Table 1 heading: “Prevenance of Weight Regain” → “Prevalence of Weight Regain.”
Table 2 heading: “Row and Cooked Food” → “Raw and Cooked Food.”
Line 185: “met the definition of significant weight regain” → “met the study definition of weight regain.” (remove “significant” redundancy).
Discussion
Line 266: “post-surgery weight management outcome” → “post-surgery weight management outcomes.”
Line 269–270: “Such findings are counterintuitive and are most likely explained” → “These counterintuitive findings are most likely explained…”
Line 311: “engage in "grazing" or frequent snacking” → “engage in grazing, or frequent snacking.”
Conclusion
Line 350–351: “Although unexpected associations were observed between certain foods and obesity status” → “Although unexpected associations were observed between certain foods and obesity.”
Author Response
Reviwer 1:
- Line 20: Replace “81.8% experienced significant weight regain, defined as a ≥20% increase from their nadir weight” → “81.8% experienced weight regain, defined as a ≥20% increase from nadir weight.” (remove “significant” as you already define it).
Author Response:
Done with many thanks
- Line 23–24: “linked to reduced obesity risk” → “associated with lower odds of obesity.” (consistent with statistical reporting).
Author Response:
Done with many thanks
Introduction
- Line 37: “Weight regains after bariatric surgery has become a well-known phenomenon” → “Weight regain after bariatric surgery is a well-recognized phenomenon.”
Author Response:
Done with many thanks
- Line 80–82: “their influenced of weight regain” → “their influence on weight regain.”
Author Response:
Done with many thanks
Methods
- Line 88: “Sharik Participants databased” → “Sharik participant database.”
Author Response:
Done with many thanks
- Line 114: Remove double period → “… outcomes.”
Author Response:
Done with many thanks
- Line 121–122: “traditional dishes such as kabsah (rice with meat or chicken) and shawarma” → “traditional dishes such as kabsah (rice with meat or chicken) and shawarma.” (add period).
Author Response:
Done with many thanks
Results
- Table 1 heading: “Prevenance of Weight Regain” → “Prevalence of Weight Regain.”
Author Response:
Done with many thanks
- Table 2 heading: “Row and Cooked Food” → “Raw and Cooked Food.”
Author Response:
Done with many thanks
- Line 185: “met the definition of significant weight regain” → “met the study definition of weight regain.” (remove “significant” redundancy).
Author Response:
Done with many thanks
Discussion
- Line 266: “post-surgery weight management outcome” → “post-surgery weight management outcomes.”
Author Response:
Done with many thanks
- Line 269–270: “Such findings are counterintuitive and are most likely explained” → “These counterintuitive findings are most likely explained…”
Author Response:
Done with many thanks
- Line 311: “engage in "grazing" or frequent snacking” → “engage in grazing, or frequent snacking.”
Author Response:
Done with many thanks
Conclusion
- Line 350–351: “Although unexpected associations were observed between certain foods and obesity status” → “Although unexpected associations were observed between certain foods and obesity.”
Author Response:
Done with many thanks
Reviewer 2 Report
Comments and Suggestions for Authors
Authors have modified adequately my comments. Congratulations